# Prevalence and Determinants of Vitamin D Deficiency in 1825 Cape Town Primary Schoolchildren: A Cross-Sectional Study

**DOI:** 10.3390/nu14061263

**Published:** 2022-03-16

**Authors:** Keren Middelkoop, Neil Walker, Justine Stewart, Carmen Delport, David A. Jolliffe, James Nuttall, Anna K. Coussens, Celeste E. Naude, Jonathan C. Y. Tang, William D. Fraser, Robert J. Wilkinson, Linda-Gail Bekker, Adrian R. Martineau

**Affiliations:** 1Desmond Tutu HIV Centre, Institute of Infectious Disease & Molecular Medicine, University of Cape Town, Cape Town 7925, South Africa; justine.stewart@hiv-research.org.za (J.S.); carmen.delport@hiv-research.org.za (C.D.); linda-gail.bekker@hiv-research.org.za (L.-G.B.); 2Department of Medicine, University of Cape Town, Cape Town 7925, South Africa; 3Wolfson Institute of Population Health, Queen Mary University of London, London E1 2AB, UK; neil.walker@qmul.ac.uk (N.W.); d.a.jolliffe@qmul.ac.uk (D.A.J.); 4Paediatric Infectious Diseases Unit, Red Cross War Memorial Children’s Hospital, Cape Town 7700, South Africa; james.nuttall@uct.ac.za; 5Department of Paediatrics and Child Health, University of Cape Town, Cape Town 7700, South Africa; 6Wellcome Centre for Infectious Diseases Research in Africa, Institute of Infectious Disease and Molecular Medicine, University of Cape Town, Cape Town 7925, South Africa; coussens.a@wehi.edu.au (A.K.C.); robert.wilkinson@uct.ac.za (R.J.W.); 7Infectious Diseases and Immune Defence Division, Walter and Eliza Hall Institute, Parkville, VIC 3052, Australia; 8Centre for Evidence-Based Health Care, Faculty of Medicine and Health Sciences, Stellenbosch University, Cape Town 7505, South Africa; cenaude@sun.ac.za; 9Norwich Medical School, University of East Anglia, Norwich Research Park, Norwich NR4 7TJ, UK; jonathan.tang@uea.ac.uk (J.C.Y.T.); w.fraser@uea.ac.uk (W.D.F.); 10Departments of Clinical Biochemistry and Endocrinology, Norfolk and Norwich University Hospitals Trust, Norwich NR4 7UY, UK; 11The Francis Crick Institute, Midland Road, London NW1 1AT, UK; 12Department of Infectious Diseases, Imperial College London, London W12 0NN, UK; 13Blizard Institute, Barts and The London School of Medicine and Dentistry, Queen Mary University of London, London E1 4NS, UK

**Keywords:** vitamin D, prevalence, South Africa, children, cross-sectional

## Abstract

Vitamin D deficiency (25-hydroxyvitamin D[25(OH)D] <50 nmol/L) is common among adults in Cape Town, South Africa, but studies investigating vitamin D status of children in this setting are lacking. We conducted a cross-sectional study to determine the prevalence and determinants of vitamin D deficiency in 1825 Cape Town schoolchildren aged 6–11 years. Prevalence of vitamin D deficiency was 7.6% (95% Confidence Interval [CI] 6.5% to 8.9%). Determinants of vitamin D deficiency included month of sampling (adjusted odds ratio [aOR] for July–September vs. January–March 10.69, 95% CI 5.02 to 22.77; aOR for October–December vs. January–March 6.73, 95% CI 2.82 to 16.08), older age (aOR 1.25 per increasing year, 95% CI: 1.01–1.53) and higher body mass index (BMI; aOR 1.24 per unit increase in BMI-for-age Z-score, 95% CI: 1.03–1.49). In a subset of 370 participants in whom parathyroid hormone (PTH) concentrations were measured; these were inversely related to serum 25(OH)D concentrations (*p* < 0.001). However, no association between participants with hyperparathyroidism (PTH >6.9 pmol/L) and vitamin D deficiency was seen (*p* = 0.42). In conclusion, we report that season is the major determinant of vitamin D status among Cape Town primary schoolchildren, with prevalence of vitamin D deficiency ranging from 1.4% in January–March to 22.8% in July–September.

## 1. Introduction

Vitamin D is a micronutrient present in some foods but largely produced in the skin following exposure to solar ultraviolet B radiation [1]. The importance of vitamin D in maintaining bone health is well established, and this micronutrient plays a critical role in bone accretion in childhood [2]. There is increasing evidence that vitamin D also plays a regulatory role in the immune system, affecting protection against acute respiratory infections [3,4], tuberculosis [5,6] and asthma exacerbations [3,7]. In addition vitamin D has been hypothesized to influence growth [8], neurodevelopment and cognitive function [9,10] and age of puberty [11]. Despite the evident importance of vitamin D for child health, there remains a paucity of data on the prevalence of vitamin D deficiency in children in Africa, and notably in southern Africa.

Across a small number of studies, high prevalences (30–45%) of vitamin D deficiency (serum 25-hydroxyvitamin D [25(OH)D] < 50 nmol/L) have been reported in north African countries in children and adolescents spanning 5 to 18 years of age [12,13,14,15]. Risk factors identified for vitamin D deficiency in these settings include older age [12], increased body fat [15], darker skin [12], lower dietary intake of vitamin D and calcium [12,14], lower socio-economic status [12,15] and reduced sun exposure [12,15,16]. On the equator, a study among children in a feeding scheme in Kenya (average age 5 years) reported <1% prevalence of vitamin D deficiency [17]. A study in Uganda reported a prevalence of 25% in children aged 18 months to 12 years [18]. In South Africa, there are no recent, published community surveys of vitamin D status among young children. In 2011, a longitudinal birth cohort in the Gauteng Province (latitude ~26° S) reported that 8% of Black children were vitamin D deficient at 10 years of age [19]. More recently, a bone health case control study reported 7% prevalence of vitamin D deficiency among 59 black pre-adolescent South African control children in the same province [20]. South Africa spans a wide range of latitudes, from sub-tropical regions in the north to the temperate regions in the South where seasonal variation in vitamin D status is more evident [21] but fewer data on prevalence of vitamin D deficiency in children are available.

We addressed this gap in knowledge and understanding of vitamin D status among children in South Africa by investigating the prevalence and determinants of vitamin D deficiency in a cohort of primary school children living in Cape Town, South Africa (latitude ~34° S) screened for participation in a clinical trial of vitamin D supplementation to prevent TB infection (clinicaltrials.gov ref NCT02880982).

## 2. Materials and Methods

The study was based in the Mitchells Plain/Klipfontein sub-district of Cape Town. This sub-district, which is home to almost 1 million people, is an underserved, multi disease burdened, peri-urban area. Residents are predominantly African Black and colored (mixed race) individuals, and the majority have either isiXhosa or Afrikaans as their home language. The communities in this sub-district are historically and currently disadvantaged, unemployment is high (>30%) and the monthly income for the majority of households less than 3200 South African Rand (ZAR; ± $205) [22]. Schools in this sub-district were identified by an internet search of all primary schools within a 10-kilometer radius of the study site. All these schools were approached to participate and as recruitment numbers required, the recruitment radius was expanded. Of the 28 schools approached, 23 schools participated in the study.

Eligible children were aged 6–11 years and enrolled in Grades 1–4 of participating public primary schools for whom parental/guardian consent and child assent were both obtained. Children were excluded from the study if they had clinical evidence of rickets; if they had any chronic illness other than asthma; if they used any regular medication other than asthma treatment; if they had taken supplemental vitamin D at a dose >400 IU daily or equivalent in the previous month; if they had received previous treatment for latent TB infection (LTBI) or active TB disease; or if they were not expected to remain in the study area for the duration of follow-up in the trial (3 years).

Following engagement with relevant school bodies, potentially eligible children at 23 public primary schools were identified through school administrative systems and class lists. Children were provided with a letter for their parent or legal guardian, inviting them to meet with study staff to discuss the study. Parents and legal guardians who indicated an interest had a home-based face-to-face meeting scheduled. Following written consent from parents/legal guardians, an interviewer confirmed participant eligibility and administered a questionnaire collecting sociodemographic information as well as details of potential determinants of their child’s baseline vitamin D status.

After completion of the parent visit informed assent was sought from the potentially eligible participants on the school premises. Assented children were screened for final eligibility assessment, including signs of symptomatic vitamin D deficiency. Eligible and assenting children had blood drawn into serum-separating tubes (SST) to determine serum concentrations of 25(OH)D and, in a subset of children screened for a nested sub-study, parathyroid hormone (PTH) concentrations.

### 2.1. Laboratory Methods

Serum concentrations of 25(OH)D were determined using liquid chromatography tandem mass spectrometry (LC-MS/MS) as previously described [23]. In brief, the method quantified 25-hydroxycholecalciferol (25[OH]D_3_) and 25-hydroxyergocalciferol (25[OH]D_2_) simultaneously. Both 25(OH)D_3_ and 25(OH)D_2_ were calibrated using standard reference material SRM972a from the National Institute of Science and Technology (NIST), and showed linearity between 0 and 200 nmol/L. The inter/intra-assay coefficient of variation (CV) across the assay range was ≤9%, and the lower limit of quantification (LLoQ) was 0.1 nmol/L. The assay showed <6% accuracy bias against NIST reference method on the vitamin D external quality assessment (DEQAS) scheme (http://www.deqas.org/; accessed on 1 September 2019). Intact PTH was analyzed on the COBAS platform (Roche Diagnostics, Mannheim, Germany) using an electrochemiluminescence immunoassay according to manufacturer’s instructions. The inter-assay CV was ≤3.8% across the analytic range of 1.2–5000 pg/mL.

### 2.2. Statistical Analysis

Statistical analysis was limited to those participants for whom a serum 25(OH)D concentration was available.

As 25(OH)D_2_ was undetectable (i.e., <10 nmol) in all samples, analysis was based on 25(OH)D_3_ results only. A concentration of <50 nmol/L for 25(OH)D was used as the cut-off to define vitamin D deficiency [2]. Body mass index (BMI) was available on the sub-set of participants who were enrolled in the primary cohort study. BMI-for-age Z-scores for individuals were calculated using the WHO 2007 Z-Score calculator shiny app [24] using the sex, age, height and weight of the participant at baseline. Hyperparathyroidism was defined as PTH concentration >6.9 pmol/L.

Predictive variables were tested for association with serum 25(OH)D concentrations in a series of univariate regressions. Two regression models were considered with (i) 25(OH)D as the response in a standard linear model with normal error and (ii) response dichotomized by whether 25(OH)D was ≥ vs. <50 nmol/L in a logistic regression model. Any of the independent variables with a p-value of less than 0.1 in univariable analysis was entered into a “final” multiple regression model so that inference was based on adjustment for important confounders. Correlation between the distribution of 25(OH)D concentrations versus PTH concentration was assessed using Spearman’s rank test, and univariate logistic regression was used to test for association between participants with hyperparathyroidism and vitamin D deficiency. As BMI data were only available for participants who subsequently enrolled onto the ViDiKids trial, multivariate models were also developed excluding the BMI-for-age Z-score variable.

Two-way scatter plots were generated to illustrate the relationship between 25(OH)D concentration and selected predictors of vitamin D status. In each case, a least-squares line of best fit was added. In the case of screening date, this was fit as a sinusoidal curve to reflect evident cyclicity; the sine and cosine of the proportion of calendar year elapsed (in radians) on screening date were calculated and fitted simultaneously in a linear model on 25(OH)D; the model coefficients were used to generate the predictive line. A scatterplot was produced to show the relationship between baseline serum 25(OH)D and serum PTH. In order to assess potential local trends in this relationship, a cubic spline was fit using the Stata default of five knots and superimposed on the plot.

Statistical analysis was performed using Stata software (Version 17.0; StataCorp, College Station, Texas, United States). GraphPad Prism software (version 9.1.0, GraphPad, San Diego, California, United States) was used to generate scatter plots of Serum concentrations of 25(OH)D by age, BMI and date of sampling.

### 2.3. Ethics

This study was approved by the University of Cape Town Faculty of Health Sciences Human Research Ethics Committee (Ref: 796/2015) and the London School of Hygiene and Tropical Medicine Observational/Interventions Research Ethics Committee (Ref: 7450-2). The trial was registered on the South African National Clinical Trials Register (DOH-27-0916-5527) and ClinicalTrials.gov (https://clinicaltrials.gov/ct2/show/NCT02880982; accessed on 25 January 2022). Parents or legal guardians were required to provide written informed consent and all participants provided written assent.

## 3. Results

Of 3108 parents visited by study staff, 151 (5%) declined consent for their child to participate, 72 (2%) were not the parent or legal guardian, and 15 (<1%) did not have mental capacity to consent. At the consenting visit, 18 (1%) children were deemed ineligible due to age or grade, previous treatment of LTBI or TB disease (*n* = 54.2%), chronic illness including suspected or known HIV (*n* = 21.1%), long-term use of non-asthma medication or vitamin D supplements (*n* = 9, <1%) and plans to move out of the study area in the next 3 years (*n* = 8, <1%). More than one ineligibility criterion was noted in several children (Figure 1). Of the 2782 eligible children whose parents consented for them to be approached for assent, 2355 (85%) assented to participate in the study, while 13 (<1%) children declined participation, 25 (1%) had aged out of eligibility prior to assent, 25 (1%) relocated outside of the study area and 20 (1%) had parents who withdrew consent or were not contactable for reconsenting on updated informed consent documents. The team were unable to contact and assent 341 (12%) children whose parents had consented for them to take part prior to the close of enrolment. Of those assented, blood was not taken from 84 (4%) children due to refusal or difficult phlebotomy, and a further eight due to other ineligibilities for the primary study (Figure 1). Among the 2263 remaining participants serum concentrations of 25(OH)D were available for 1825 (81%) participants.

Cohort baseline characteristics are outlined in Table 1 and Table 2. The median age of participants was 9.3 years, 53% were female and 98% identified as isiXhosa. Recruitment occurred from April 2017 to May 2019 and 69% of participants were recruited in the first 6 months of any given calendar year (January–June). Parents tended to have low levels of education (83% had primary school level only), and the median monthly household income was 1200 ZAR (interquartile range [IQR] 760 to 2500). The mean 25(OH)D concentration at baseline was 73.5 nmol/L (standard deviation [SD] 17.6) and 7.6% (95% Confidence Interval [CI]; 6.5% to 8.9%) had 25(OH)D concentration <50 nmol/L. More specifically, 0.05% (95% CI: 0.01% to 0,39%) had 25(OH)D concentration <25 nmol/L, 7.6% (95% CI: 6.43% to 8.87%) between 25 and 49.9 nmol/L, 48.3% (46.0% to 50.6%) between 50 and 74.9 nmol/L and 44.1% (95% CI: 41.8% to 46.3%) ≥75 nmol/L.

Table 1 presents results of linear regression analysis of potential determinants of 25(OH)D concentration analyzed as a continuous variable, while Table 2 presents results of logistic regression analysis of potential determinants of vitamin D deficiency expressed as a binary categorical variable (25(OH)D <50 nmol/L vs. ≥50 nmol/L). In both the linear and logistic regression models, female sex, increasing age and sampling in April–December were independently associated with lower vitamin D status. Higher BMI-for-age Z-score (> +3) was significantly associated with lower 25(OH)D concentrations as a continuous variable, and met the cut-off for inclusion in multivariate model in logistic regression (*p* = 0.08). The distribution of 25(OH)D concentrations by age, BMI-for-age Z-score and date of sampling are illustrated in Figure 2.

In univariate linear regression analysis, informal housing was associated with lower 25(OH)D concentrations analyzed as a continuous dependent variable (*p* = 0.002), but not with the binary outcome of 25(OH)D concentrations <50 nmol/L in logistic regression (*p* = 0.56). Conversely, a trend towards 25(OH)D concentrations >50nmolL was noted with higher margarine consumption (*p* = 0.07), but this was not noted in the linear regression (*p* = 0.58).

Age, sex, season, type of housing and BMI-for-age Z-score were included in the multiple regression models. In the final linear regression model for determinants of 25(OH)D concentrations (Table 1), female sex, increasing age, sampling in April to December (autumn through spring) and higher BMI-for-age Z-score were all independently associated with lower 25(OH)D concentrations. In the final logistic regression model (Table 2), increasing age, IsiXhosa ethnicity, sampling in winter or spring and higher BMI-for-age Z-score were all independently associated with 25(OH)D concentrations <50 nmol/L (Table 2). Across both multiple linear and logistic regression models, sampling from July to September was most strongly associated with lower 25(OH)D concentrations (β coefficient for comparison with sampling in January–March −21.3, 95% CI: −23.7 to −18.9; and OR 11.2, 95% CI: 5.3 to 24.0, respectively) followed by sampling in October to December (β coefficient for comparison with January–March −16.8, 95% CI: −19.9 to −13.8; and OR 6.7, 95% CI: 2.8 to 15.0, respectively). Neither housing type nor margarine consumption remained significantly associated with low 25(OH)D concentrations in multiple regression models. Excluding BMI-for-age Z-score from the multivariate models did not substantively change the outcomes in the linear multivariate regression models (Appendix A). In the logistic regression models, excluding BMI strengthened the association between risk of vitamin D deficiency and female sex (*p* = 0.02) and attenuated the association between isiXhosa ethnicity and risk of vitamin D deficiency to statistical non-significance (*p* = 0.64) (Appendix A).

Parathyroid hormone concentrations were available for a subset of 370 participants. The median PTH concentration was 3.5 pmol/L (IQR 2.7 to 4.5) and 3.5% (95% CI, 2.0 to 6.1) had PTH concentration >6.9 pmol/L (the upper limit of normal range). PTH concentrations were inversely related to serum 25(OH)D concentrations (*p* < 0.001) (Figure 3). Table 3 reports the proportions of participants with elevated vs. normal PTH concentrations, stratified by serum 25(OH)D concentration: of note, 3/55 (5.5%) participants with 25(OH)D <50 nmol/L had PTH concentrations >6.9 pmol/L, compared with 10/315 (3.2%) of those with 25(OH)D concentrations ≥50 nmol/L (*p* = 0.42, Fisher’s exact test).

## 4. Discussion

In this cross-sectional study, we report that 7.6% of young schoolchildren in a socio-economically deprived community in Cape Town, were vitamin D deficient at the 50 nmol/L 25(OH)D threshold. The key risk factor for 25(OH)D deficiency was season of testing, with sampling July to December (winter and spring) associated with increased risk of vitamin D deficiency as compared to the first quarter of the year (summer). Increasing age, higher BMI and female sex were also independently associated with lower 25(OH)D concentrations in multivariable analyses.

The prevalence of Vitamin D deficiency identified in this study is lower than the >40% prevalence reported in North African countries [12,13,15], but is consistent with reports of 7–8% prevalence of 25(OH)D <50 nmol/L among children in more northern regions of South Africa including Johannesburg/Tshwane (previously Pretoria) (latitude ~26°S) [19,20]. However, the prevalence is much lower than might be expected on the basis of a report showing a 63% prevalence of vitamin deficiency among adults in the Western Cape [25]; this difference may reflect reduced sun exposure in adults vs. children, arising as a consequence of spending more time indoors e.g., due to indoor employment. While there is some variability across studies with regards to key determinants of vitamin D status, the importance of exposure to sunlight or seasonal variation is a consistent finding in non-equatorial settings [12,15,19]. The inverse association between 25(OH)D concentrations and BMI is also well recognized in the literature [2,15,19]. While such an association was not reported in a study conducted in Gauteng Province, South Africa, it should be noted that that study’s small sample size (*n* = 59) may have limited its power to detect an association [20].

Diet was not identified as an important determinant of 25(OH)D status in this cohort. Interestingly, increasing margarine consumption did trend towards a protective association with 25(OH)D concentrations in univariate logistic modelling. This would be biologically plausible, as many brands of margarine are fortified with vitamin D_2_ in South Africa, and further, bread is one of the four main foods contributing to total energy and macronutrient intake in children aged 6–9 years in the Western Cape [26]. However, this association was not sustained in multivariable models. While some studies have noted a role of diet in vitamin D status [12,14], our finding of the absence of diet as a determinant of 25(OH)D concentrations is consistent with the less significant role of diet as a major contributor to vitamin D status noted in settings with high sunlight exposure [1,2]. This contrasts with findings of studies conducted in settings where sunlight is more limited, where higher intake of vitamin D-containing foods such as eggs has been reported to associate independently with higher 25(OH)D concentrations [27]. In our study, female sex was associated with lower 25(OH)D concentrations, consistent with some other studies [28,29]. Multivariate models excluding BMI strengthened the positive association between female sex and low 25(OH)D concentrations, as expected given the strong correlation between these variables.

The strengths of this study include the wide range of determinants of 25(OH)D status explored, as well as the fact that sampling was performed throughout the year, in order to capture the full range of seasonal variation [16,25]. Furthermore, 25(OH)D concentrations were determined using LC-MS/MS—the gold standard method for determining concentrations of this metabolite [29]. The laboratory that assayed 25(OH)D concentrations for this study participates in, and meets the certification criteria of, the internationally recognized Vitamin D External Quality Assessment Scheme (DEQAS). The study was well powered with a large sample size, and the results are likely generalizable to the wider population of isiXhosa-speaking children living in the Western Cape, who form one of the dominant population groups in the area.

Our study also has some limitations. Firstly, the study sample is not representative of the ethnic diversity of the City of Cape Town where Black Africans comprise 39% of the population [30]. Ethnic differences in the prevalence of vitamin D deficiency have been reported in South Africa [19], and therefore our results may not be generalizable to different ethnic groups living in the Western Cape. Similarly, the results may not be generalizable to children outside of the study age range. In addition, these data were not obtained from a cross-sectional survey of the general population, but from a population who were being screened for participation in a clinical trial. However, we highlight that among those screened no one was excluded due to clinical evidence of 25(OH)D deficiency, and very few (2%) were excluded due to previous treatment for LTBI or TB disease, or HIV infection. Weight and height data were only available for those who were enrolled into the clinical trial, and BMI data were therefore missing for participants who were excluded from the trial due to a positive interferon gamma release assay (IGRA) test result at screening.

## 5. Conclusions

We report a 7.6% prevalence (95% CI: 6.5–8.9) of vitamin D deficiency at the 50 nmol/L 25(OH)D threshold among young Black African school children living in a socio-economically disadvantaged area of Cape Town. This prevalence is similar to that reported among children living in more northern areas of South Africa, but lower than anticipated based on the prevalence of vitamin D deficiency previously reported among adults in the Western Cape. Season of sampling, most likely related to changes in sun exposure, was the most powerful determinant of vitamin D status in this population. The only other modifiable determinant of low vitamin D status identified was higher BMI, which highlights the importance of nutritional and fitness education and programs in schools.

## Figures and Tables

**Figure 1 nutrients-14-01263-f001:**
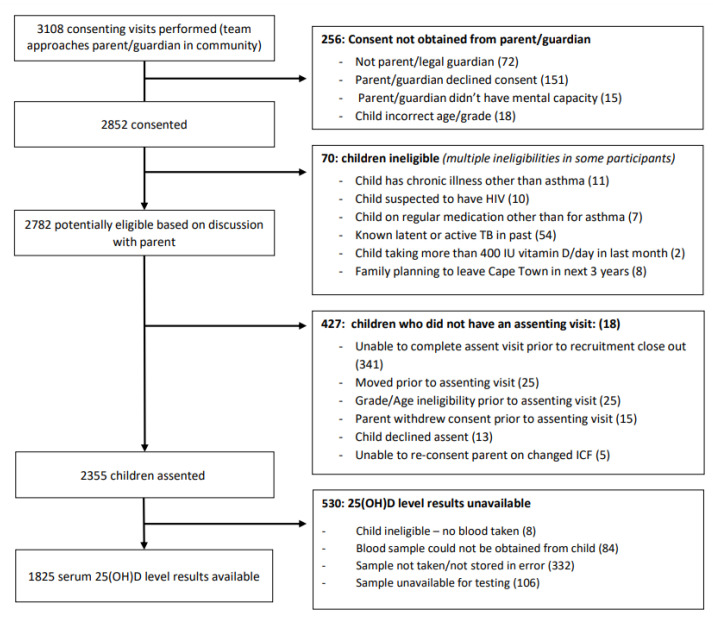
Consort diagram for screening and enrolment.

**Figure 2 nutrients-14-01263-f002:**
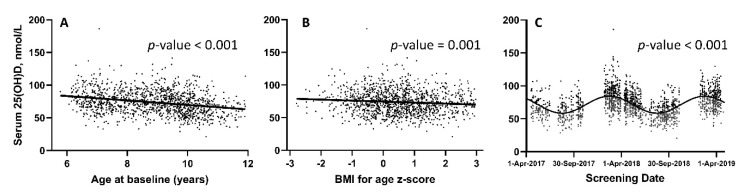
Serum concentrations of 25(OH)D by age (**A**), BMI (**B**) and date of sampling (**C**). Regression lines are fitted to (**A**) [slope: −3.42 (95% CI: −4.07 to −2.76)] and (**B**) [slope: −1.54 (95% CI: −2.43 to −0.65)]; best fit line (sinusoidal curve) fitted to (**C**).

**Figure 3 nutrients-14-01263-f003:**
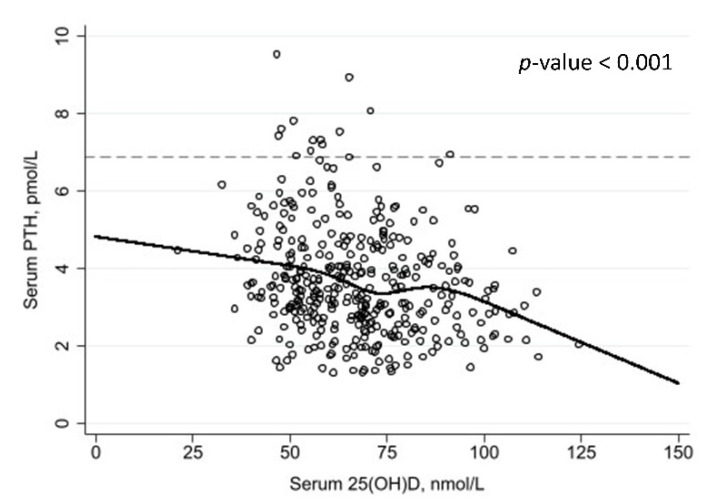
Serum concentrations of 25(OH)D vs. parathyroid hormone concentrations. Restricted cubic spline fitted (5 knots); reference line: 6.9 pmol/L.

**Table 1 nutrients-14-01263-t001:** Determinants of mean 25(OH)D concentration: univariate and multiple linear regression analysis.

			Univariate Regression		Multiple Regression	
	*N*	Mean 25(OH)D Concentration at Baseline, nmol/L (s.d.)	Mean Difference (95% CI)	*p*-Value	Mean Difference (95% CI)	*p*-Value
**Sex** *** Male Female						
	854	74.7 (17.6)	0 (reference)	0.01	0 (reference)	<0.001
	969	72.5 (17.4)	−2.17 (−3.79 to −0.56)		−3.02 (−4.63 to −1.41)	
**Age at assent and screening (years)** ***						
6	168	80.4 (16.2)	−3.31 (−3.89 to −2.74)	<0.001	−1.42 (−2.09 to −0.75)	<0.001
7	281	77.3 (18.1)				
8	346	75.9 (16.1)				
9	503	73.9 (17.6)				
10	396	68.4 (16.5)				
11+	108	62.1 (16.0)				
**Ethnicity** *** isiXhosa Colored Other *						
	1756	73.4 (17.6)	0 (reference)	0.98	N/A	
	15	73.5 (12.7)	+0.09 (−8.84 to +9.01)			
	17	72.7 (16.1)	−0.75 (−9.14 to +7.64)			
**Season screened**						
January–March	657	83.0 (16.8)	0 (reference)	< 0.001	0 (reference)	< 0.001
April–June	605	74.5 (14.6)	−8.48 (−10.15 to −6.81)		−8.18 (−10.12 to −6.24)	
July–September	360	59.0 (12.9)	−23.93 (−25.87 to −21.98)		−21.33 (−23.73 to −18.93)	
October–December	193	65.9 (14.6)	−17.01 (−19.43 to −14.58)		−16.83 (−19.85 to −13.80)	
**Parental Level of Education** **						
No formal education	49	75.2 (20.3)	+0.68 (−1.33 to +2.69)	0.51	N/A	
Primary school level	1504	73.3 (17.5)				
Secondary school or higher	271	74.5 (17.2)				
**Household income** (1000 ZAR per month)						
Q1 (0 – 0.76)	416	72.1 (17.7)	+0.094 (−0.19 to +0.38)	0.52	N/A	
Q2 (0.78 – 1.2)	407	73.5 (16.9)				
Q3 (1.3 – 2.5)	547	73.8 (17.3)				
Q4 (2.6 – 45)	454	74.5 (18.2)				
**Type of accommodation** ***						
Brick	948	72.3 (17.1)	0 (reference)	0.002	0 (reference)	0.13
Informal	876	74.8 (17.9)	+2.54 (+0.94 to +4.15)		+1.25 (−0.38 to +2.87)	
**Egg Consumption**						
Never	27	71.6 (14.7)	−0.66 (−1.60 to +0.27)	0.17	N/A	
1–4 times in past month	210	74.8 (18.3)				
1–2 times per week	850	73.5 (18.1)				
3–5 times per week	565	74.1 (16.7)				
Every day/almost every day	172	70.5 (16.6)				
**Liver Consumption**						
Never	86	69.7 (14.2)	−0.75 (−1.67 to +0.18)	0.12	N/A	
1–4 times in past month	790	75.2 (18.2)				
1–2 times per week	655	72.5 (17.2)				
3–5 times per week	241	72.5 (17.0)				
Every day/almost every day	52	72.1 (17.4)				
**Red Meat Consumption**						
Never	112	73.8 (16.2)	−0.67 (−1.71 to +0.37)	0.21	N/A	
1–4 times in past month	1009	73.8 (18.3)				
1–2 times per week	521	73.4 (16.9)				
3–5 times per week	166	72.0 (15.6)				
Every day/almost every day	16	70.9 (16.6)				
**Fish Consumption**						
Never	47	76.0 (15.3)	+0.12 (−0.69 to +0.83)	0.75	N/A	
Less than once a month	204	71.9 (16.2)				
1–4 times in past month	441	72.6 (17.0)				
1–2 times per week	723	74.8 (18.8)				
3–5 times per week	284	72.5 (17.0)				
Every day/almost every day	125	73.0 (15.6)				
**Margarine Consumption**						
Never	17	72.4 (17.0)	+0.28 (−0.69 to +1.24)	0.58	N/A	
1–4 times in past month	59	72.5 (18.7)				
1–2 times per week	127	71.6 (17.2)				
3–5 times per week	462	74.4 (19.3)				
Every day/almost every day	1159	73.5 (16.8)				
**Anyone in household smokes cigarettes indoors**						
No	1541	73.7 (17.7)	0 (reference)	0.32	N/A	
Yes	283	72.6 (16.6)	−1.14 (−3.36 to +1.09)			
**BMI-for-age Z-score** ***						
<−2	1155	74.0 (17.7)	−1.21 (−1.92 to −0.51) ^†^	0.001	−0.95 (−1.55 to −0.34) ^†^	0.002
+2 to +3	131	72.6 (15.6)				
>+3	45	69.3 (16.9)				
**Time outside during daylight**						
<1 h per day	6	53.7 (20.0)	+1.76 (−0.81 to +4.34) ^‡^	0.18	N/A	
1–2 h per day	173	73.1 (17.6)				
>2 h per day	1644	73.6 (17.5)				
**Amount of skin not covered when outdoors**						
Just face and hands	166	72.3 (17.8)	+1.79 (−0.60 to +4.18) ^#^	0.14	N/A	
Arms/legs as well as face and hands	1608	73.5 (17.5)				
Often no shirt, as well as face, hands, arms, legs	49	77.1 (18.2)				
**Sunscreen use**						
Usually	41	73.2 (13.4)	+0.29 (−5.15 to +5.73)	0.92	N/A	
Not usually	1782	73.5 (17.6)				

* Other included: Sotho (10), Venda (2), Shona (2), Swazi (1), Malawian (1) and Zimbabwean (1). ** Highest level of education of at least one parent. *** Missing data points: gender for three participants; age, 23; ethnicity, 37; parental level of education, one; type of housing, one; BMI-for-age Z-score,1. ^†^ B-Coefficient represents reduction in 25(OH)D concentration (nmol/L) per unit increase of Z-score. ^‡^ B-Coefficient represents increase in 25(OH)D concentration (nmol/L) per additional extra hour spent outdoors during daylight hours. ^#^ B-Coefficient represents increase in 25(OH)D concentration (nmol/L) per category of increasing skin exposure when outdoors.

**Table 2 nutrients-14-01263-t002:** Determinants of vitamin D deficiency (25[OH]D <50 nmol/L): univariate and multivariate logistic regression analysis.

			Univariate Regression		Multiple Regression	
	*N*	Percentage with 25(OH)D <50 nmol/L-N (%)	Crude Odds Ratio (95% CI)	*p*-Value	Adjusted Odds Ratio (95% CI)	*p*-Value
**Sex** *** Male Female						
	854	52 (6.1%)	1 (reference)	0.03	1 (reference)	0.09
	969	85 (8.8%)	1.48 (1.04 to 2.12		1.49 (0.94 to 2.36)	
**Age at assent and screening (years)** ***						
6	168	4 (2.4%)	1.70 (1.46 to 1.98)	<0.001	1.25 (1.01 to 1.53)	0.04
7	281	8 (2.9%)				
8	346	18 (5.2%)				
9	503	35 (7.0%)				
10	396	48 (12.1%)				
11+	108	22 (20.4%)				
**Ethnicity** isiXhosa Colored Other *						
	1756	135 (7.7%)	1 (reference)	0.08	1 (reference)	<0.001
	15	0 (0%)	-		1 (-)	
	17	1 (5.9%)	0.75 (0.10 to 5.70)		1.07 (0.11 to 9.91)	
**Season screened**						
January–March	657	9 (1.4%)	1 (reference)	<0.001	1 (reference)	<0.001
April–June	605	22 (3.6%)	2.72 (1.24 to 5.95)		1.59 (0.67 to 3.79)	
July–September	360	82 (22.8%)	21.24 (10.52 to 42.87)		10.69 (5.02 to 22.77)	
October–December	193	23 (11.9%)	9.74 (4.43 to 21.44)		6.73 (2.82 to 16.08)	
**Parental Level of Education** **						
No formal education	49	4 (8.2%)	0.74 (0.47 to 1.17)	0.20	N/A	
Primary school level	1504	119 (7.9%)				
Secondary school or higher	271	15 (5.5%)				
**Household income** (ZAR 1000 per month)						
Q1 (0 – 0.76)	416	37 (8.9%)	0.98 (0.91 to 1.05)	0.55	N/A	
Q2 (0.78 – 1.2)	407	30 (7.4%)				
Q3 (1.3 – 2.5)	547	42 (7.7%)				
Q4 (2.6 – 45)	454	29 (6.4%)				
**Type of accommodation** ***						
Brick	948	75 (7.9%)	1 (reference)	0.56	N/A	
Informal	876	63 (7.2%)	0.90 (0.64 to 1.28)			
**Egg Consumption**						
Never	27	1 (3.7%)	1.11 (0.91 to 1.36)	0.29	N/A	
1–4 times in past month	210	13 (6.2%)				
1–2 times per week	850	66 (7.8%)				
3–5 times per week	565	42 (7.4%)				
Every day/almost every day	172	16 (9.3%)				
**Liver Consumption**						
Never	86	6 (7.0%)	1.13 (0.93 to 1.37)	0.23	N/A	
1–4 times in past month	790	50 (6.3%)				
1–2 times per week	655	58 (8.9%)				
3–5 times per week	241	21 (8.7%)				
Every day/almost every day	52	3 (5.8%)				
**Red Meat Consumption**						
Never	112	7 (6.3%)	0.91 (0.73 to 1.15)	0.45	N/A	
1–4 times in past month	1009	84 (8.3%)				
1–2 times per week	521	34 (6.5%)				
3–5 times per week	166	13 (7.8%)				
Every day/almost every day	16	0 (0%)				
**Fish Consumption**						
Never	47	4 (8.5%)	0.94 (0.81 to 1.09)	0.41	N/A	
Less than once a month	204	14 (6.9%)				
1–4 times in past month	441	37 (8.4%)				
1–2 times per week	723	57 (7.9%)				
3–5 times per week	284	20 (7.0%)				
Every day/almost every day	125	6 (4.8%)				
**Margarine Consumption**						
Never	17	0 (0%)	0.84 (0.70 to 1.01)	0.07	0.80 (0.62 to 1.04)	0.10
1–4 times in past month	59	8 (13.6%)				
1–2 times per week	127	15 (11.8%)				
3–5 times per week	462	36 (7.8%)				
Every day/almost every day	1159	79 (6.8%)				
**Anyone in household smokes cigarettes indoors**						
No	1541	117 (7.6%)	1 (reference))	0.92	N/A	
Yes	283	21 (7.4%)	0.98 (0.60 to 1.58)			
**BMI-for-age Z-score *****						
<−2	1155	81 (7.0%)	1.15 (0.99 to 1.35) ^†^	0.08	1.24 (1.03 to 1.49) ^†^	
+2 to +3	131	7 (5.3%)				
>+3	45	7 (15.2%)				
**Time outside during daylight**						
<hour per day	6	2 (33.3%)	0.70 (0.43 to 1.15) ^‡^	0.16	N/A	
1–2 hours per day	173	15 (8.7%)				
>2 hours per day	1644	121 (7.4%)				
**Amount of skin not covered when outdoors**						
Just face and hands	166	16 (9.6%)	0.72 (0.44 to 1.16) # ^#^	0.18	N/A	
Arms/legs as well as face and hands	1608	120 (7.5%)				
Often no shirt, as well as face, hands, arms, legs	49	2 (4.1%)				
**Time outside during daylight**						
<hour per day	6	2 (33.3%)	0.70 (0.43 to 1.15) ^‡^	0.16	N/A	
1–2 hours per day	173	15 (8.7%)				
>2 hours per day	1644	121 (7.4%)				
**Sunscreen use**						
Usually	41	0 (0%)			N/A	
Not usually	1782	138 (7.8%)				

* Other included: Sotho (10), Venda (2), Shona (2), Swazi (1), Malawian (1) and Zimbabwean (1). ** Highest level of education of at least one parent. *** Missing data points: gender for two participants; age, 23; ethnicity, 37; parental level of education, one; type of housing, one; BMI (Z-score),1. ^†^ B-Coefficient represents reduction in 25(OH)D concentration (nmol/L) per unit increase of Z-score. ^‡^ B-Coefficient represents increase in 25(OH)D concentration (nmol/L) per additional extra hour spent outdoors during daylight hours. ^#^ B-Coefficient represents increase in 25(OH)D concentration (nmol/L) per category of increasing skin exposure when outdoors.

**Table 3 nutrients-14-01263-t003:** PTH concentrations above and below the upper limit of normal range (6.9 pmol/L) stratified by serum 25(OH)D concentrations.

	Serum 25(OH)D, nmol/L			
	<25.0	25.0–49.9	50.0–74.9	≥75.0
PTH ≤ 6.9 pmol/L	1 (100.0%)	51 (94.4%)	199 (95.7%)	106 (99.1%)
PTH > 6.9 pmol/L	0 (0.0%)	3 (5.6%)	9 (4.3%)	1 (0.9%)
**Total**	**1**	**54**	**208**	**107**

## Data Availability

An anonymized dataset is available from ARM upon reasonable request.

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
