# Peer review of "Prevalence and Determinants of Vitamin D Deficiency in 1825 Cape Town Primary Schoolchildren: A Cross-Sectional Study"

_nutrients, 2022, doi:10.3390/nu14061263_

Round 1
Reviewer 1 Report
The manuscript entitled “Prevalence and determinants of Vitamin D deficiency in 1,825 Cape Town primary schoolchildren: a cross-sectional study” presents interesting data about vitamin D status in children living in Cape Town, South Africa. The authors report that 7.6% of young schoolchildren in a socio-economically deprived community in Cape Town and the risk factors for 25(OH)D deficiency were season (winter and spring), increasing age, higher BMI and female sex.
I have the following remarks:
Is it possible to show individual measurements for 25(OH)D3 and 25(OH)D2 for typical results? I think that will make the paper more informative.
Is it possible to show the cubic spline curve between PTH and 25(OH)D in relation to Figure 3, as in many previous reports?
Author Response
- Is it possible to show individual measurements for 25(OH)D3 and 25(OH)D2 for typical results? I think that will make the paper more informative.
All 25(OH)D2 concentrations were classified as undetectable (i.e. <10 nmol) and thus did not contribute significantly to vitamin D status in this group. Therefore analysis was based on 25(OH)D3 results. This has been clarified in the manuscript:
“Statistical analysis was limited to those participants for whom a serum 25(OH)D concentration was available. As all 25(OH)D2 concentrations were reported as undetectable (i.e. <10 nmol), analysis was based on 25(OH)D3 results. “
- Is it possible to show the cubic spline curve between PTH and 25(OH)D in relation to Figure 3, as in many previous reports?
Thank you for this suggestion. We have used the cubic spline curve in Figure 3, as suggested. This is addressed in the methods section and in Figure 3 in the results.
“A scatterplot was produced to show the relationship between baseline serum 25(OH)D and serum PTH. In order to assess potential local trends in this relationship, a cubic spline was fit using the Stata default of 5 knots and superimposed on the plot”
Reviewer 2 Report
Vitamin D deficiency is common among adults in Cape Town, South Africa, but studies investigating vitamin D status of children in this setting are lacking. This study conducted a cross-sectional study to determine the prevalence and determinants of vitamin D deficiency in 1825 Cape Town schoolchildren aged 6-11 years. Prevalence of vitamin D deficiency was 7.6% (95% Confidence Interval [CI] 6.5% to 8.9%). It does fill the data gap of vitamin D deficiency in children in South Africa. I have some concerns that should be addressed by the authors. Given below are my detailed comments.
- What is the sampling method used in this study? How many schools in the region match the research criteria? What is the basis for the investigation of selected schools in this study? Is it sampling by division or cluster? Please clarify the specific method in the manuscript.
- The purpose of this study is investigating the prevalence and determinants of vitamin D deficiency in a cohort of primary school children. However, the study only measured the concentration of PTH. Why not detect and analyze the relationship between other serological indicators and vitamin D levels? Please explain.
- As a cross-sectional study, the sample size of this study is too small, and it is doubtful whether it is representative.
Author Response
- What is the sampling method used in this study? How many schools in the region match the research criteria? What is the basis for the investigation of selected schools in this study? Is it sampling by division or cluster? Please clarify the specific method in the manuscript.
The study location was chosen based on the high TB infection and disease incidence, as well as anticipated high levels of Vitamin D deficiency. Schools were chosen based on a google search of all primary schools within a 10km radius of the study site. All these schools were approached to participate, and as recruitment numbers required, we expanded the recruitment radius. Of the 28 schools approached, 23 schools participated. This information has been added to the methods section.
“Schools in this sub-district were identified by an internet search of all primary schools within a 10 kilometre radius of the study site. All these schools were approached to participate and as recruitment numbers required, the recruitment radius was expanded. Of the 28 schools approached, 23 schools participated in the study.”
- The purpose of this study is investigating the prevalence and determinants of vitamin D deficiency in a cohort of primary school children. However, the study only measured the concentration of PTH. Why not detect and analyze the relationship between other serological indicators and vitamin D levels? Please explain.
PTH was investigated in a subset of participants as it is the single best physiologic indicator of the impact of vitamin D status on calcium homeostasis – the primary function of Vitamin D in the body. It is not clear which other serological indicators the reviewer has in mind, but regrettably further laboratory analyses cannot be conducted for the current manuscript as this would require a protocol amendment and applications for further funding.
- As a cross-sectional study, the sample size of this study is too small, and it is doubtful whether it is representative.
To our knowledge, the study is by some way the largest to investigate vitamin D status of South African primary school children in recent times [see literature review by Norval et al 2016] - 385 largest samples size in children; White et al 2019 - sample size of 84]. While the sample size provides sufficient power to detect statistically significant associations between key determinants of vitamin D status and 25(OH)D levels, this paper reports on the baseline results of a cohort study powered to detect impact of Vitamin D supplementation on TB infection rates in primary school children (as noted in the Methods and limitations sections). Therefore, the effects investigated and presented were not powered for in the original sample size calculation, but a sample size of 1,825 was considered adequate in this context.
We acknowledge that results cannot be generalised to other age groups, or to children of other ethnic groups. We have noted and clarified in the discussion that “….and therefore our results may not be generalisable to different ethnic groups living in the Western Cape. Similarly, the results may not be generalisable to children outside of the study age range.”
However, the study is representative of Black children living in Cape Flats, as we achieved good coverage of local schools, and we do not claim that our findings are generalisable to populations other than those studied.
Round 2
Reviewer 2 Report
Accept in present form.
Author Response
Manuscript has been reviewed for minor spelling/grammar checks.